

# Robust weather-adaptive postprocessing using MOS random forests

Thomas Muschinski[1,3], Georg J. Mayr[1], Achim Zeileis[2], and Thorsten Simon[2]

[1]Department of Atmospheric and Cryospheric Sciences, Universität Innsbruck, Innsbruck, Austria
[2]Department of Statistics, Universität Innsbruck, Innsbruck, Austria
[3]Department of Economics, Statistical Methods and Econometrics, Karlsruhe Institute of Technology, Karlsruhe, Germany

**Correspondence:** Thomas Muschinski (Thomas.Muschinski@kit.edu)

**Abstract.** Physical numerical weather prediction models have biases and miscalibrations that can depend on the weather situation, which makes it difficult to postprocess them effectively using the traditional model output statistics (MOS) framework based on parametric regression models. Consequently, much recent work has focused on using flexible machine learning methods that are able to take additional weather-related predictors into account during postprocessing, beyond the forecast of the variable of interest only. Some of these methods have achieved impressive results, but they typically require significantly

more training data than traditional MOS and are less straightforward to implement and interpret.

We propose MOS random forests, a new postprocessing method that avoids these problems by fusing traditional MOS with a powerful ML method called random forests to estimate "weather-adapted" MOS coefficients from a set of predictors. Since the assumed parametric base model contains valuable prior knowledge, much smaller training data sizes are required to obtain

skillful forecasts and model results are easy to interpret. MOS forests are straightforward to implement and typically work well, even with no or very little hyperparameter tuning. For the difficult task of postprocessing daily precipitation sums in complex terrain, MOS forests outperform reference machine learning methods at most of the stations considered. Additionally, they are highly robust to changes in the data size and work well even when less than a hundred observations are available for training.

## 1 Introduction

Although physically-based numerical weather predictions (NWPs) have made significant improvements in recent decades (Bauer et al., 2015), statistical postprocessing is still necessary to correct systematic errors in the forecasts and accurately quantify their uncertainty (Vannitsem et al., 2021). The popular *model output statistics* (MOS) framework introduced by Glahn and Lowry (1972) postprocesses NWPs using linear regressions between historical observations and their corresponding predictions. Since then, the idea behind MOS has been extended to ensemble-postprocessing (EMOS) using more flexible regression

models that allow for heteroscedastic forecast errors (NGR, Gneiting et al., 2005) or non-Gaussian responses (e.g., Scheuerer, 2014; Simon et al., 2019).

Postprocessing with MOS or EMOS is intuitive and can work well, but requires a dataset that is both sufficiently large to allow for stable estimation of model coefficients and homogeneous enough so these may be assumed constant. To obtain such a dataset, it is standard practice to estimate separate MOS for different atmospheric quantities, locations and lead times.

Seasonal changes in predictability can be accounted for using time-adaptive MOS that employ sliding window training schemes



(Gneiting et al., 2005) or by replacing constant model coefficients with cyclical functions of the day of the year (Lang et al., 2020). This approach also works with other univariate predictors such as altitude (Schoenach et al., 2020).

Weather-adaptive postprocessing – i.e., allowing biases and miscalibrations of the NWP model to depend on the weather situation – is necessary to obtain optimal forecast performance, but is made complicated by the large number of potentially
relevant atmospheric variables whose interactions are unknown or poorly understood. It is possible to include such additional predictors in a MOS model by using selection procedures based on expert knowledge (Stauffer et al., 2017b) or gradient boosting (Messner et al., 2017), but this requires that the interactions are either ignored or parameterized a priori.

Machine learning (ML) methods have become increasingly popular postprocessing tools in recent years because they are well-suited to deal with this high-dimensional predictor space (Schulz and Lerch, 2022). Neural networks (NNs), for example,
have been used in parametric distributional regressions similar to EMOS (Rasp and Lerch, 2018) and semi-parametric quantile function regressions based on Bernstein polynomials (Bremnes, 2020). The predictive skill of NNs can be impressive, but they typically require combining data from many different stations to effectively train the model. Purely local (station-wise) ML-based postprocessing is often performed using random forests, which generally assume either a parametric distribution for the response (Schlosser et al., 2019) or predict a collection of specified quantiles (Taillardat et al., 2016), although combinations of
the two have been employed as well (Taillardat et al., 2019). Random forests have the advantage of being straightforward to implement, but in the MOS applications they must learn the typically approximately linear relationship between the observations and the model outputs by combining many (highly nonlinear) step functions.

MOS random forests fuse traditional and ML-based postprocessing by first assuming an appropriate parametric MOS model and then adapting its coefficients to the weather situation at hand using random forests. The split variables and corresponding
split points in the individual trees of a MOS forest are not selected based on properties of the response variable directly (e.g., their mean, quantiles, or other parameters) as done in quantile forests or distributional forests. Instead the splits are chosen based on changes in the *MOS coefficients* of the assumed model, which may reflect either changes in the marginal distribution of the response (e.g., captured by intercepts) or changes in the dependence on the model outputs (e.g., captured by slopes). The predictor space is thus partitioned to ensure homogeneity with respect to the MOS coefficients, meaning that a single model
with constant coefficients can be assumed to work well in each corresponding subsample of the data. In order to decrease variance and allow for smooth dependencies, a MOS forest combines the partitions from many different MOS trees grown using bootstrapped or subsampled data (Breiman, 1996) and only random subsets of predictor variables for splitting at each node (Breiman, 2001). Weather-adapted MOS coefficients predicted by the MOS forest can then be interpreted and used for postprocessing in the usual way.

A detailed description of MOS forests can be found in Sec. 2. In the following Sec. 3, MOS forests and reference methods are used to postprocess ensemble predictions of daily-precipitation sums in complex terrain. The results of this real-world application are presented in Sec. 4. Strengths and limitations of the proposed method are discussed in Sec. 5 and summarizing remarks conclude the paper in Sec. 6.



## 2 MOS random forests

MOS forests adapt the regression coefficients of an assumed (non-adaptive) base MOS to some set of additional atmospheric variables that characterize the current weather situation. Thus, it is first necessary to choose a suitable base MOS for the specific postprocessing task at hand (Sec. 2.1). Subsequently, individual *MOS trees* are grown from this base MOS using model-based recursive partitioning algorithms which seek to identify "homogeneous weather partitions" of the predictor space within the tree's terminal nodes (Sec. 2.2). Individual MOS trees already allow for weather-adaptive postprocessing but can only approximate smooth effects through step functions with many splits. To better capture smooth effects and improve predictive performance, MOS forests therefore combine the partitions from not just one, but many different MOS trees learned on random subsamples of the full data, yielding the final weather adapted MOS (Sec. 2.3). This model can then be used for postprocessing "as usual".

### 2.1 Choosing a base MOS

The goal of MOS is to improve upon the quality of physical NWP models by identifying their "weather-related statistics" using regression models trained on historical observations and corresponding predictions (Glahn and Lowry, 1972). Since MOS was first introduced fifty years ago, there have been substantial changes in both (i) what is meant by "weather-related statistics" in the context of MOS and (ii) the flexibility of the regression methods used to identify these.

In the simplest case – with a single (deterministic) forecast for an atmospheric quantity and forecast errors that may be assumed Gaussian – systematic biases in the NWPs can be identified using a classical linear regression. A classical example is to regress observed temperatures $y$ on the corresponding temperature predictions $x$:

$$E(y \mid x) = \beta_0 + \beta_1 \cdot x. \tag{1}$$

MOS coefficients $\beta_0$ and $\beta_1$ then describe how the temperature forecast from the physical model should be corrected to better match real world observations. For the ideal case of an NWP with no systematic biases, these values would be $\beta_0 = 0$ and $\beta_1 = 1$. In the classical linear model, coefficients are estimated by miniziming the sum of the squared errors (OLS) on some set of training data, which is equivalent to minimizing the root mean square error (RMSE) of the residuals.

This simple postprocessing model not only allows biases in the NWP to be corrected, but also implicitly estimates the uncertainty of the postprocessed forecast. Namely, if $y$ can be assumed to follow a Gaussian distribution conditionally on $x$, the minimum RMSE obtained during model estimation is an estimate of the standard deviation $\sigma$ of the forecast distribution and Eq. 1 may be rewritten as

$$y \sim \mathcal{N}(\mu, \sigma^2), \quad \text{where} \quad \mu = \beta_0 + \beta_1 \cdot x. \tag{2}$$

Typically though, weather forecasts do not have constant uncertainty and many atmospheric variables do not follow Gaussian distributions, even conditionally. To allow for more flexibility in postprocessing, modern implementations of MOS therefore often employ distributional regressions (Kneib et al., 2021), also known as generalized additive models for location, scale



and shape (GAMLSS, Rigby and Stasinopoulos, 2005). In distributional regression, the observation $y$ can follow some other parametric distribution and all parameters (not just the mean) of this distribution are modeled on appropriate predictors derived from the NWP (ensemble).

Typically, coefficients of distributional regression models are estimated by maximizing the log-likelihood $\ell$ of the distributional parameters given the observations or by minimizing the continuous ranked probability score (CRPS). One prominent example in the postprocessing literature is nonhomogeneous Gaussian regression (NGR), also known as EMOS (Gneiting et al., 2005), where the parameters $\mu$ and $\sigma$ in Eq. 2 are modeled on the mean and spread of an NWP ensemble, respectively. Other examples include truncated Gaussian and generalized extreme value response distributions to forecast wind speeds (Thorarinsdottir and Gneiting, 2010; Lerch and Thorarinsdottir, 2013) and censored and shifted gamma distributions to forecast precipitation (Baran and Nemoda, 2016).

In the subsequent sections we therefore assume that the base MOS for $y$ explained by $x$ uses some parametric model with likelihood $\ell((y,x),\boldsymbol{\theta})$ and $r$-dimensional parameter vector $\boldsymbol{\theta}$ that is estimated through likelihood-maximization:

$$\hat{\boldsymbol{\theta}} = \underset{\boldsymbol{\theta}}{\operatorname{argmax}} \sum_{i=1}^{N} \ell((y_i, x_i), \boldsymbol{\theta}), \tag{3}$$

In the example from Eq. 2 the likelihood is Gaussian with parameter vector $\boldsymbol{\theta} = (\beta_0, \beta_1, \sigma)$ but other distributions, like the ones from the previous paragraph, could be used in the same way.

## 2.2 Growing individual MOS trees

In order to adapt the coefficients of the base MOS chosen in Sec. 2.1 to some additional weather-related predictors $z_1, z_2, \ldots z_k$, a single MOS tree partitions the predictor space $Z_1 \times Z_2 \times \cdots \times Z_k$ into disjoint subsets that can each be considered "homogeneous weather situations for the purpose of NWP postprocessing" – i.e., where constant MOS coefficients work well. It is grown using model-based recursive partitioning algorithms (Zeileis et al., 2008; Seibold et al., 2018) according to the following steps:

**Step 1: Estimate coefficients of the base MOS**

MOS coefficients $\boldsymbol{\theta}$ are estimated through likelihood maximization on the $i = 1, \ldots, N$ observations $y_i$ and corresponding predictions $x_i$ in the dataset. This is done by solving the first-order condition

$$\sum_{i=1}^{N} s((y_i, x_i), \boldsymbol{\theta}) = \mathbf{0}, \tag{4}$$

where

$$s((y_i, x_i), \boldsymbol{\theta}) = \left( \frac{\partial \ell((y_i, x_i), \boldsymbol{\theta})}{\partial \theta_1}, \ldots, \frac{\partial \ell((y_i, x_i), \boldsymbol{\theta})}{\partial \theta_r} \right)^{\top} \tag{5}$$

contains the partial derivatives of the log-likelihood with respect to each coefficient – i.e., the model scores – evaluated at the $i$-th observation pair $(y_i, x_i)$.





**Step 2: Select the splitting variable**

Scores with respect to each coefficient are again computed at all observations (Eq. 5) and evaluated at the estimated coefficients $\hat{\boldsymbol{\theta}} = (\hat{\theta}_1, \ldots, \hat{\theta}_r)$ from Step 1. Since the estimated coefficients were obtained using Eq. 4, each score vector has a mean of zero. If the single MOS with constant coefficients fits well, the scores for each observation should randomly fluctuate around zero. On the other hand, systematic departures of the scores from zero along some of the variables in $\boldsymbol{z}$ suggest that predictions can be improved by splitting the data and estimating separate postprocessing models on the two resulting subsamples. Whether or

not the scores fluctuate randomly or depend on one of the weather-related predictors can be assessed using an independence test between the scores and each of the variables in $\boldsymbol{z}$. If there is a significant dependence with respect to at least one of the variables, then the most significant variable is selected for splitting. To account for assessing multiple variables from $\boldsymbol{z}$, a Bonferroni correction for multiple testing is employed.

**Step 3: Identify the optimal split point**

Once the splitting variable $z_j$ has been selected, an exhaustive search is performed over all possible split points to identify the partition that improves the log-likelihood the most. For numerical splitting variables, up to $2 \cdot (N - 1)$ different MOS are estimated in this step – separate models in both subsamples for each of the $N - 1$ possible split points. The number of possible split points (and thus estimated models) decreases for each tie among the realizations of $z_j$. For unordered categorical splitting variables, the number of possible split points is equal to the number of ways in which the different categories can be divided

into two subgroups, and thus increases exponentially with the number of distinct categories.

**Repeat previous steps**

The three steps described above split a dataset of size $N$ into two disjoint subsamples that are then each postprocessed using a separate MOS. In order to grow a MOS tree, these steps are repeated for each subsample until a stopping criterion has been reached. The terminal nodes of a MOS tree (i.e., those nodes that are not split any further) contain disjoint subsamples of the

full data that correspond to different *homogeneous* weather situations for postprocessing with MOS (Figs. 1 and 2).

Coefficients $\beta_0, \beta_1, \gamma_0$ in each terminal node are obtained through likelihood-maximization on the corresponding subsample. Note that this can also be understood as a weighted estimated using the full data, where weights are either zero or one, indicating whether or not the respective observation is in the subsample of interest. In the following Sec. 2.3, this idea is extended to use weights that may change smoothly (rather than abruptly) between zero and one. This can express the degree of similarity (with

respect to MOS coefficients) between some new weather situation and those historical weather situations in the training data.

**2.3 Obtaining weather-adapted coefficients from a random forest of MOS trees**

Individual MOS trees grown according to Sec. 2.2 are easy to understand and interpret (see Sec. 4.1), but can be sensitive to small changes in the data and may have a suboptimal fit if the model parameters change smoothly with the weather situation variables. To solve this problem and improve out-of-sample predictive skill, a MOS forest combines partitions from many

 

different trees grown on bootstrap-aggregated (bagged) data and using only a randomly chosen subset of the atmospheric
variables in $\boldsymbol{z}$ for splitting at each node.

Given a MOS forest with $T$ trees and $P^t$ partitions in each tree $t$, MOS coefficients are adapted to a new weather situation
$\boldsymbol{z}^\star \in Z_1 \times Z_2 \times \cdots \times Z_r$ by maximizing the likelihood of the base MOS on the full training data as in Eq. 3,

$$\hat{\boldsymbol{\theta}}(\boldsymbol{z}^\star) = \operatorname*{argmax}_{\boldsymbol{\theta}} \sum_{i=1}^{N} w(\boldsymbol{z}^\star, \boldsymbol{z}^i) \cdot \ell((y_i, x_i), \boldsymbol{\theta}), \tag{6}$$

but with observations $(y_i, x_i)$ weighted according to

$$w(\boldsymbol{z}^\star, \boldsymbol{z}^i) = \frac{1}{T} \sum_{t=1}^{T} \sum_{p=1}^{P^t} \frac{\mathbb{1}((\boldsymbol{z}^\star \in \mathcal{P}_p^t) \wedge (\boldsymbol{z}^i \in \mathcal{P}_p^t))}{|\mathcal{P}_p^t|}. \tag{7}$$

These weights thus capture how similar the new weather situation $\boldsymbol{z}^\star$ is to any of the historical weather situations $\boldsymbol{z}^i$ from the
training data, by computing how often they end up in the same "homogenous weather partition" from the different trees in the
forest. Thus, they characterize their similarity with respect to the MOS coefficients.

By using partitions from many different trees to estimate the weather-adapted MOS, model coefficients are not restricted to
a discrete number of unique values at most equal to the number of terminal nodes (as can be seen with estimates for $\sigma$ from the
MOS tree of Fig. 3). Instead, coefficients are allowed to have smooth dependencies on the additional predictors and as a result
predictions are more stable (see estimates for $\sigma$ from the MOS forest of Fig. 3).

The MOS coefficients $\hat{\boldsymbol{\theta}}(\boldsymbol{z}^\star)$ that have been adapted to the new weather situation $\boldsymbol{z}^\star$ can be used to postprocess the cor-
responding forecast $x^\star$ in the same way as coefficients obtained from a MOS tree or from the base MOS itself. That is, the
(log-transformed) probability density function for the unkown observation $y^\star$ is given by $\ell(y^\star \mid x^\star, \hat{\boldsymbol{\theta}}(\boldsymbol{z}^\star))$ and the parameters
of the response distribution are those values predicted by the MOS.

## 3 Postprocessing precipitation forecasts in complex terrain

The MOS forests described in Sec. 2 are applied to the difficult task of obtaining reliable probabilistic precipitation forecasts in
complex terrain. Individual topographical features cannot be resolved by NWP models, which means that predictions for these
locations rely heavily on subgrid scale parameterizations whose accuracy can depend on the weather situation. Postprocessing
models are trained and evaluated on the `RainTyrol` dataset described in Sec. 3.1, which contains observations of daily
precipitation sums and various ensemble-derived predictor variables that can be used for weather-adaptive postprocessing. The
exact configuration of the MOS forest for this application and a description of the three reference methods are given in Sec. 3.2.

### 3.1 Data

The `RainTyrol` dataset (Schlosser et al., 2019) is composed of observed daily precipitation sums from the Austrian National
Hydrographical Service and NWPs from the 11-member global ensemble forecast system (GEFS, Hamill et al., 2013) of the
U.S. National Oceanic and Atmospheric Administration (NOAA). Data is available at 95 different stations in Tyrol, Austria





| Model name | Forecast type | Prespecified regression model for | | Splitting variables |
| --- | --- | --- | --- | --- |
| | | Location: $\mu$ | Scale: $\log(\sigma)$ | |
| MOS forest | censored Gaussian | $\beta_0 + \beta_1 \cdot \texttt{tppow\_mean}$ | $\gamma_0$ | all, except `tppow_mean` |
| Distributional forest | censored Gaussian | | | all |
| Quantile regression forest | set of quantiles | | | all |
| EMOS | censored Gaussian | $\beta_0 + \beta_1 \cdot \texttt{tppow\_mean}$ | $\gamma_0 + \gamma_1 \cdot \log(\texttt{tppow\_sprd})$ | |

**Table 1.** Overview of methods used to postprocess precipitation forecasts from the `RainTyrol` dataset (see Sec. 3.1). In the dataset, variable names `tppow_mean` and `tppow_sprd` refer to the mean and standard deviation of the power-transformed ensemble forecasts of total precipitation, respectively.

(and surrounding border regions) for all July days between 1985 and 2012. July is a month with some of the largest precipitation amounts and variability within the year and both large scale precipitation events and local convective events occur. To reduce skewness, daily precipitation sums observed at 06 UTC (`robs`) are power transformed using a parameter value of $1/1.6$ following Stauffer et al. (2017a). The dataset contains a total of 80 different predictor variables derived from the GEFS, including the mean and standard deviation of the ensemble forecast of total precipitation (i.e., the direct predictors). A thorough description of all available predictor variables and their naming conventions can be found in Schlosser et al. (2019).

## 3.2 Methods

The ensemble forecasts described in Sec. 3.1 are postprocessed using MOS forests, two other forest-based weather-adaptive reference methods, and a non-adaptive EMOS. An overview of the methods is given in Table 1 and more details are supplied below.

### 3.2.1 MOS forests

To deal with the fact that precipitation sums are strictly nonnegative, we follow Schlosser et al. (2019) and assume a left-censored Gaussian response distribution with log-likelihood given by

$$
\ell(\mu, \sigma; y) = \begin{cases} \log(\frac{1}{\sigma} \cdot \phi(\frac{y - \mu}{\sigma})), & \text{if } y > 0 \\ \log(\Phi(\frac{-\mu}{\sigma})), & \text{if } y = 0 \end{cases}.
\tag{8}
$$

The prespecified base MOS

$$
\mu = \beta_0 + \beta_1 \cdot \texttt{tppow\_mean}, \qquad \log(\sigma) = \gamma_0,
\tag{9}
$$

linearly models the distributional mean $\mu$ on the mean of the (power-transformed) daily precipitation sums predicted by the individual ensemble members. The standard deviation of the response distribution $\sigma$ is modeled by an intercept.

MOS forests are estimated in R with the `model4you` (Seibold et al., 2019) and `crch` (Messner et al., 2016) packages, using the same hyperparameters as the distributional forests of Schlosser et al. (2019).



### 3.2.2 Distributional forests

Distributional forests (Schlosser et al., 2019) work in a similar fashion to MOS forests, but do not contain a prespecified MOS model. Instead, $\theta$ only contains the parameters of the assumed response distribution – i.e., in this case $\mu$ and $\sigma$ of a censored Gaussian – rather than the MOS coefficients. Trees are split with respect to distributional parameters rather than MOS coefficients and the forest estimates the postprocessed response distribution rather than a weather-adapted MOS. Distributional forests are estimated with the `disttree` package in R following the model configuration chosen by Schlosser et al. (2019).

### 3.2.3 Quantile regression forests

Both MOS forests and distributional forests require specifying a parametric response distribution a priori. Since this assumption may not always hold (even conditionally), a fully non-parametric method called quantile regression forests (Meinshausen and Ridgeway, 2006; Taillardat et al., 2016) is also considered. Splits are chosen with respect to the response value as in the standard random forest algorithm Breiman (2001), but the partitions are subsequently used to perform weighted quantile

regressions and generate probabilistic forecasts. In this application 99 quantiles are considered, corresponding to probabilities of $p = 0.01, 0.02, \dots 0.99$. Model estimation is performed using the `quantregForest` (Meinshausen, 2017) package in R.

### 3.2.4 EMOS

All three methods described above incorporate additional predictors using forest-based algorithms to allow for weather-adaptive postprocessing. In order to quantify the benefit that comes with this added model flexibility, a simple fully-parametric

non-adaptive EMOS is also considered:

$$\mu = \beta_0 + \beta_1 \cdot \texttt{tppow\_mean}, \qquad \log(\sigma) = \gamma_0 + \gamma_1 \cdot \log(\texttt{tppow\_sprd}). \tag{10}$$

This EMOS has the same mean model as the prespecified MOS in the MOS forest, but also linearly models $\log(\sigma)$ on the log-transformed standard deviation of the ensemble precipitation forecasts.

## 4 Results

To illustrate how postprocessing with MOS forests works in practice, first a single MOS tree is grown for the station of Axams and analyzed in Sec. 4.1. Subsequently, MOS forests are used to postprocess forecasts at all stations and their quality is evaluated in Sec. 4.2.

### 4.1 Interpreting a MOS tree

A MOS tree for Axams is grown from the first 24 years of data and visualized in Fig. 1. The first split of the tree sepa-

rates rare ($n = 23$) weather situations with very high ensemble-averaged total column liquid condensate (`tcolc_mean_mean`) from the remainder of the data. The rest of the data is then split based on the maximum temperature predicted by the





ensemble (`tmax_mean_mean`). The lower temperature branch has two subsequent splits: first based on precipitable water (`pwat_mean_max`) and then on the ensemble spread of precipitation (`tppow_sprd`). This results in three terminal nodes (nodes 5, 6, and 7). The higher temperature branch has three splits: based again on (`tcolc_mean_mean`), as well as the ensemble spreads of 500hPa temperature (`t500_sprd_min`) and precipitation (`tppow_sprd1824`). This results in four terminal nodes (nodes 11, 12, 13, and 14).

MOS models for each terminal node (i.e., distinct weather situation) are visualized in Fig. 2. The majority of observations are found in either node 5, 13, or 11. For nodes 5 and 13, the MOS are quite similar, the largest difference being that forecasts in node 13 are less certain (i.e., $\gamma_0$ is greater). In contrast, the MOS used to postprocess NWPs in node 11 is very different, with a strongly negative intercept for the mean model ($\beta_0 = -8.16$) and a high forecast uncertainty ($\gamma_0 = 1.09$). This is because node 11 contains many days where the ensemble mean is greater than zero – i.e., some ensemble members predict precipitation for Axams – although no precipitation is actually observed. To understand in what situations this happens, it is only necessary to consider the splits in Fig. 1 that lead to node 11: high maximum temperature, low column liquid condensate, and narrow ensemble spreads for minimum temperature at 500hPa and accumulated precipitation between 18 and 24 UTC.

## 4.2 Evaluating predictive skill

MOS forests are compared to the reference methods described in Sec. 3.2 by evaluating the skill of postprocessed forecasts using the widely-used continuous ranked probability score (CRPS, Matheson and Winkler, 1976; Gneiting and Raftery, 2007). To replicate a true operational scenario, all evaluations are performed out-of-sample on data that was not used to train the models. First, forecasts at Axams are evaluated using multiple replications of a randomized 7-fold cross validation (Sec. 4.2.1). At all other stations, forecasts are issued for a single hold-out fold (containing the last 4 years) and the remaining 6 folds (containing the first 24 years) are used for model training (Sec. 4.2.2). Finally, models are also trained using different amounts of data (12, 6, and 3 years) to investigate their robustness in this respect (Sec. 4.2.3).

### 4.2.1 Full cross validation at individual station

The Axams data is randomly split into 7 disjoint folds that each contain observations and NWPs from 4 different years. MOS forests and the reference postprocessing methods outlined in Sec. 3.2 are trained on 6 out of the 7 folds and then used to make predictions on the remaining fold. After 7 rounds of this, out-of-sample predictions are available for each day in the 28 years of data and used to compute an average CRPS for each method. The entire process is then repeated 10 times, each with a different random choice for the 7 folds. CRPS skill scores are computed relative to the EMOS model and visualized by boxplots in Fig. 4. MOS forests improve CRPS by more than 7% at Axams and thus perform slightly better than both the distributional forest and the quantile regression forest, which each lead to improvements of around 6%.




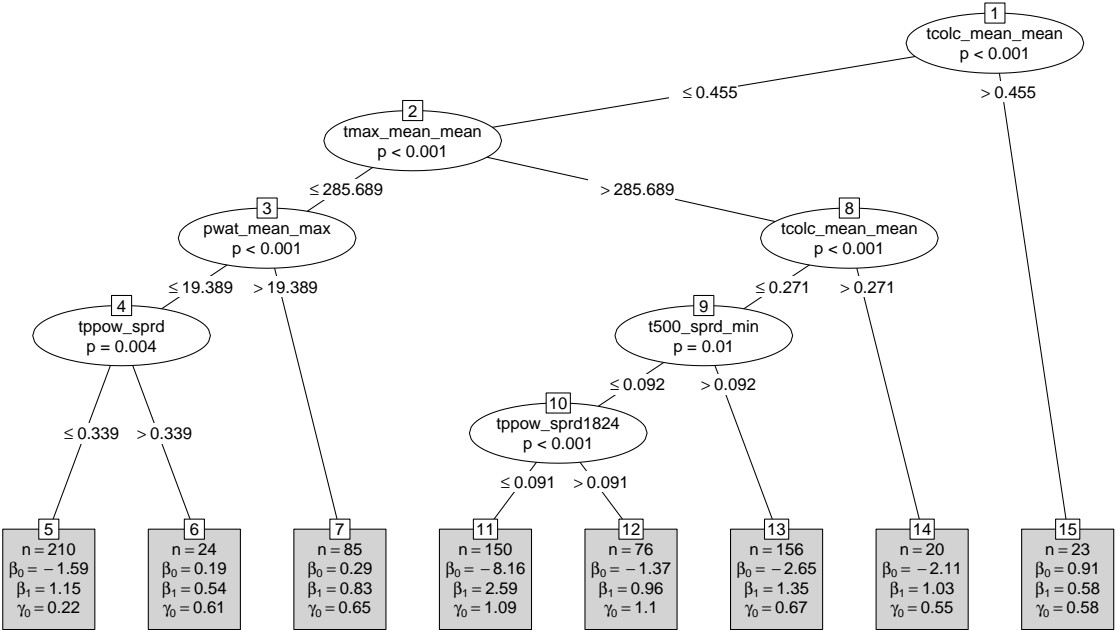

**Figure 1.** A single MOS tree estimated for Axams. Ellipses represent nodes used for splitting and contain the name of the splitting variable along with the p-value of the independence test. The corresponding split point is included in the two branches (lines) emanating from the node. Terminal nodes (which are not split again) are visualized by rectangles and contain the number of observations $n$ and estimated MOS coefficients $\beta_0, \beta_1, \gamma_0$. The models fit in each terminal node are visualized in Fig. 2.

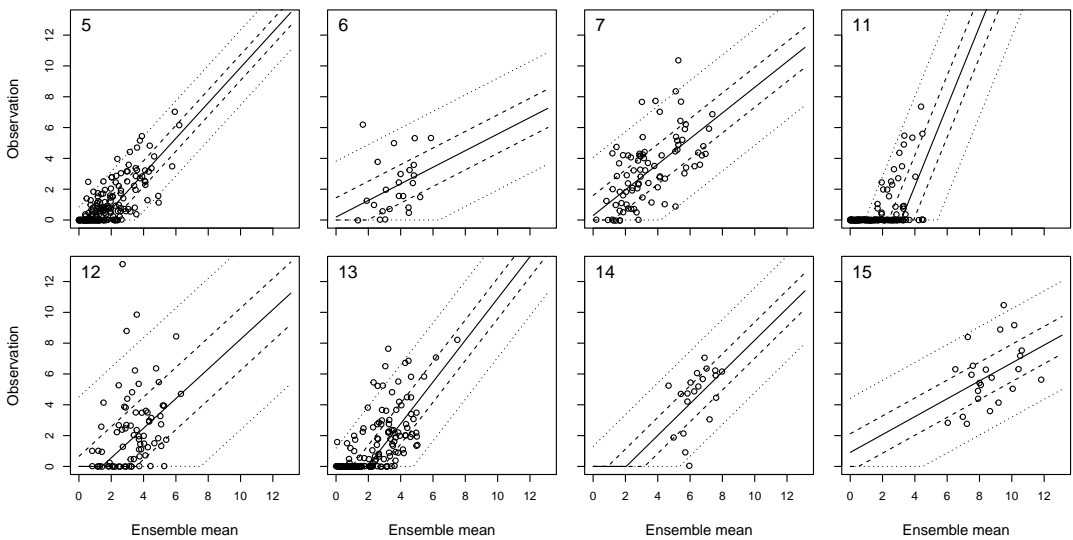

**Figure 2.** Scatter plots of observations versus ensemble mean forecasts in each terminal node of Fig. 1. Numbers identifying the nodes are included in the top left of each plot. Dashed and solid ines are quantiles corresponding to probabilities of 2.5%, 25%, 50%, 75%, and 97.5%, obtained from the MOS model fit in each node.

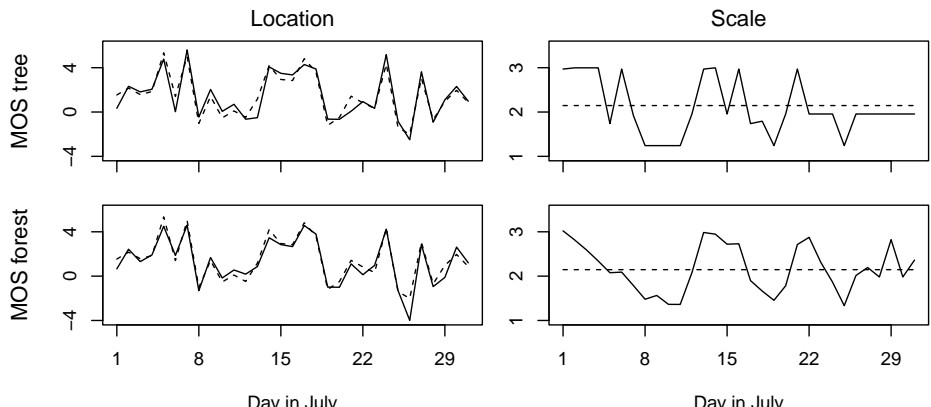

**Figure 3.** Solid lines are out-of-sample predictions for $\mu$ and $\sigma$ at Axams in July 2009, obtained from the MOS tree visualized in Figs. 1 and 2 as well as a MOS forest. Dashed lines are corresponding predictions from the base MOS (Eq. 9).

### 4.2.2 Hold out validation at all stations

To investigate predictive performance at all 95 stations, all models are trained on the first 24 years of data (1985-2008) and out-of-sample predictions are made for the last 4 years (2009-2012).

CRPS skill scores relative to EMOS are computed for each method at each station and visualized by boxplots in Fig. 4. MOS forests generally outperform the other forest-based postprocessing methods and are noticeably more robust. Distributional forests and quantile regression forests occasionally perform up to 5% worse than a basic EMOS and the quantile regression forest is outperformed by EMOS nearly 25% of the time. This is not the case for the MOS forests, which always perform at least as well as EMOS and improve the forecasts by more than 5% at 75% of the stations.

Regional differences in model performance can be seen in the map of Fig. 5. While MOS forests significantly outperform distributional forests and quantile regression forests in the northeast and southeast of the forecast region, results are less clear in the more mountainous regions further west and near the main Alpine crest. At these locations, quantile regression forests often perform slightly better.

Overall, probabilistic forecasts obtained from the MOS forests not only have a better CRPS than those obtained from the other two methods, but are also more statistically consistent with observations (i.e., calibrated). Calibration across all stations is visualized by PIT histograms for MOS forests and distributional forests and a rank histogram for the quantile regression forests (Fig. 6). Although all methods somewhat overestimate probabilities for high precipitation events, this overestimatation is much less pronounced in the MOS forests.

### 4.2.3 Sensitivity to size of training data

The methods compared above use 24 years of data for model training, but since such large datasets are not always available in postprocessing – e.g. for newly erected observational sites – the hold-out evaluations for all stations in Sec. 4.2.2 are repeated



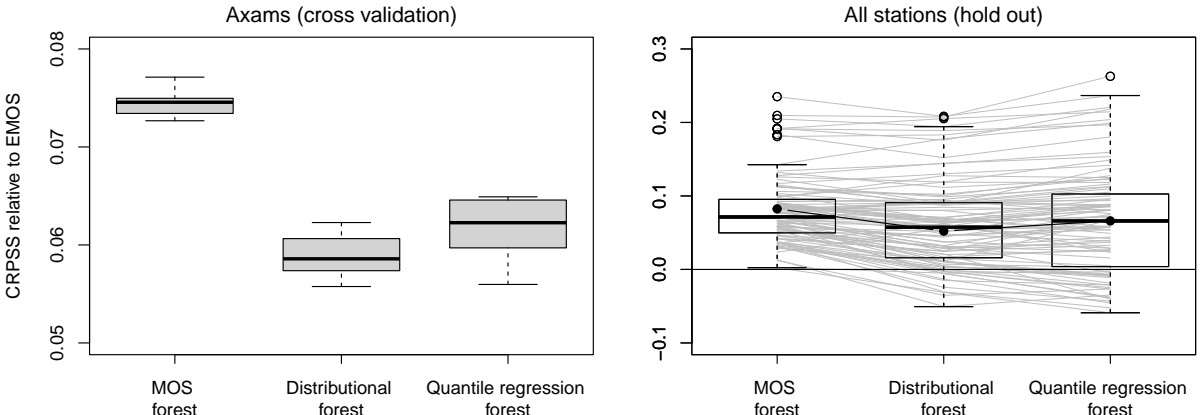

**Figure 4.** Left: CRPSS relative to EMOS at Axams based on 10 randomly chosen 7-fold cross validations. Right: CRPSS relative to EMOS at each station for the time period 2009-2012. Individual stations are connected by thin grey lines. Scores for the station of Axams are indicated by filled black circles connected by black lines.

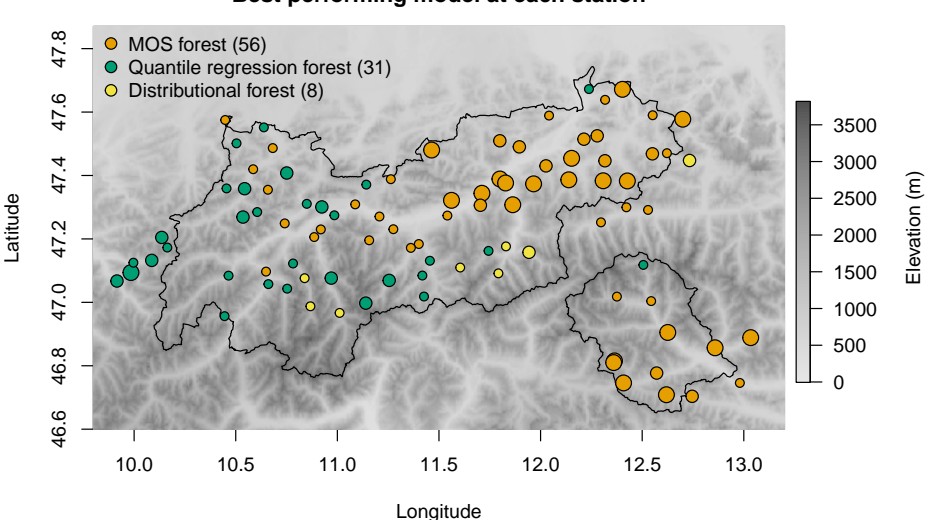

**Figure 5.** Map showing the postprocessing method that performs best at each station. Three different circle sizes (small, medium, large) are used to indicate where differences in CRPS to the second best method are less than 0.015, between 0.015 and 0.03, and more than 0.03, respectively.

using only 12, 6, and 3 years of data for training. The boxplots in Fig. 7 show that MOS forests are very robust to these changes and still perform significantly better than a non-adaptive EMOS even when trained using only 3 years of data (i.e., 93 observations). In contrast, distributional forests nearly always perform significantly worse than EMOS in such cases and have a



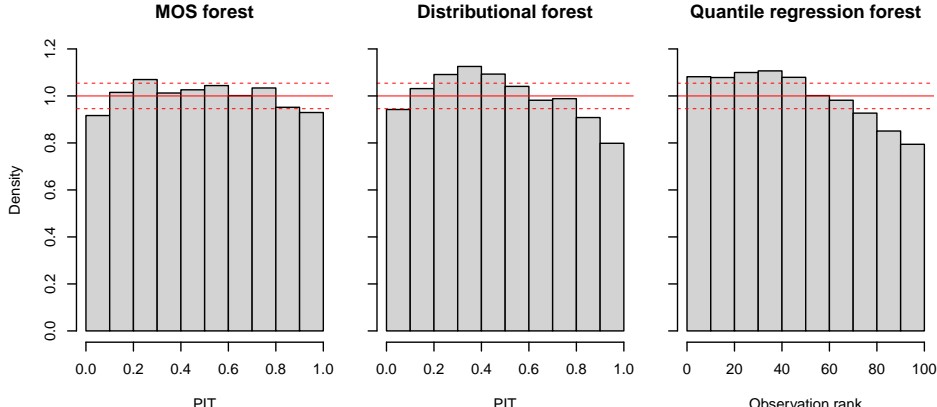

**Figure 6.** PIT histograms for MOS forests and distributional forests and rank histogram for quantile regression forests across all stations for the time period 2009-2012. Red dashed lines are the 95 % confidence intervals for a uniform distribution.

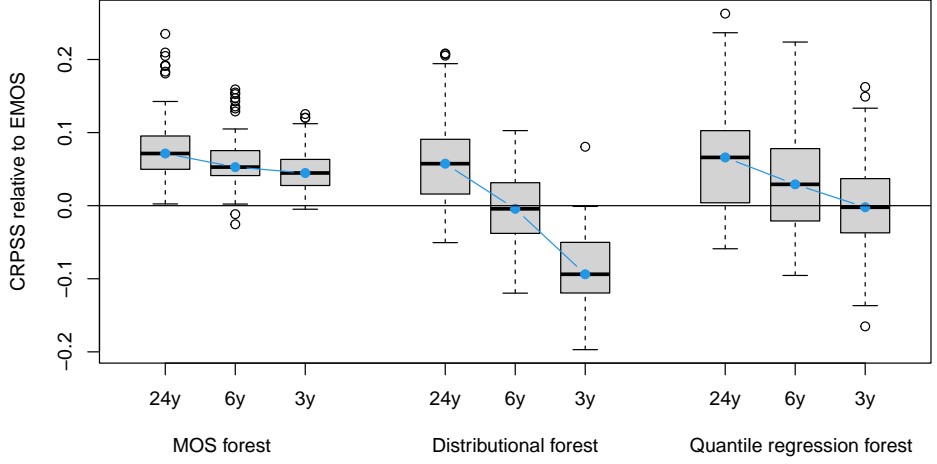

**Figure 7.** As for the hold out evaluation of all stations in Fig. 4, but with models trained on the past 24, 6 and 3 years, respectively. Blue lines highlight the influence changing data size has on the median CRPSS of each method.

median skill score of $-10\%$ across all stations. Similarly, quantile regression forests are also outperformed by the non-adaptive

EMOS at around half of the stations.

## 5    Discussion

When compared to state-of-the-art weather-adaptive postprocessing methods, MOS forests have the main advantage of being highly robust: they reliably outperform simple non-adaptive reference methods even when trained on very little data. This is possible because, unlike state-of-the-art weather-adaptive methods that treat all predictors equally and use a data-driven





approach to learn their relationships to the response, MOS forests directly incorporate prior (physically-based) knowledge about the most important relationships in the form of a parametric model. One might think that robustness is not important in our current "big data" era, but consider that NWP models are continuously updated (e.g., with improved resolutions or parameterizations) and new stations (or measurement instruments) are installed all the time. In the words of Glahn and Lowry (1972), "data samples containing numerical model output are a perishable commodity." This is still true today.

In the application considered here, MOS forests are used to postprocess NWP ensembles and separate models are estimated for each station. Without any modifications, MOS forests also offer a powerful way to obtain probabilistic forecasts from deterministic NWPs, where no predictors explicitly characterizing the forecast uncertainty are available. Similarly, MOS forests could also be employed as spatial (rather than station-wise) postprocessing models by including predictors that contain information about the individual grid points or stations within the training data. Potentially relevant variables would then include

latitude, longitude, and altitude, but also surface roughness, landcover type, or other characteristics.

Despite their many advantages, MOS forests require specifying the same two things as all other MOS models: (i) a parametric distribution for the response and (ii) models linking the parameters of that distribution with appropriate predictors derived from the NWP. Not much can be done about the first point besides trying different response distributions or transformations of the data. As for the second point, in cases where no suitable models for the distributional parameters can be specified a priori (e.g.,

if the variable observed is not a direct output of the NWP model), MOS forests have no advantage over distributional forests. In fact, MOS forests collapse to distributional forests if the assumed base MOS has intercept-only models for the parameters of the assumed response distribution.

## 6 Conclusions

Since NWPs have errors that can depend on the weather-situation, weather-adaptive postprocessing methods are necessary

to obtain optimal probabilistic forecasts. By fusing traditional (non-adaptive) and modern (weather-adaptive) postprocessing approaches, MOS forests retain the best of both worlds: a method that is flexible enough to allow for weather-adaptive postprocessing, but also robust, intuitive, and straightforward to implement. This is achieved by using random forests to adapt the regression coefficients of a prespecified parametric base MOS to a set of additional predictor variables that characterize the current weather situation. In contrast to state-of-the-art postprocessing methods, which typically directly estimate properties

of the response from these predictors, MOS forests only use them to estimate the regression coefficients of the assumed base model. As a result, they can generate skillful forecasts even when only a very limited amount of data is available for training and purely data-driven weather-adaptive methods fail to outperform a simple non-adaptive model.

*Author contributions.* TM, GJM, AZ, and TS planned the research. TM wrote the original manuscript draft and all authors subsequently reviewed and revised it.



*Competing interests.* The authors declare that they have no competing interests.

*Acknowledgements.* This project was funded by the Austrian Science Fund (FWF, grant no. P 31836). TM was also supported by the Doktoratsstipendium of Universität Innsbruck.



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
