# Peer review of "Robust weather-adaptive postprocessing using MOS random forests"

_EGUsphere, 2023_

## Author Comment (AC1)

**Responses to Reviewer 1**

*In this paper, a new postprocessing approach for the correction of forecasts' systematic errors and a quantification of their uncertainty are presented. The method is based on the use of random forests. The method also uses a local regression approach to adjust the parameters that describe the dependence of the conditional probability density function of the observations on the forecast. The method is validated using daily precipitation data and forecasts from the GFSE.*

*The paper is well written, the methodology is, in general, well described, and the results are evaluated in a statistically robust way. Below, I indicate some suggestions or comments pointing out some places in which the discussion can be clarified or in which more information should be provided.*

Thank you for taking the time to read our paper and for your constructive comments! Please find our point-by-point responses below.

*L40: This statement is unclear. In the case of random forests, the resulting fit would be smoothed out by the ensemble, so the steps would not be so obvious in the output.*

Thanks for the comment, we have clarified this statement now. You are correct that the forest ensemble will smooth the step functions to a certain degree, but the result will still be a rougher approximation of a linear function compared to estimating this directly. This should be conveyed more clearly now.

*Section 2.2. Step 2. In this section, an independence test is used to identify possible dependencies between predictors and the model parameters $\theta$. There are different variables and parameters. Could the authors elaborate more on how the split-variable is selected? Is it the one with the lowest p-value with respect to any of the parameters? Which is the independence test used in this implementation?*

The split-variable is chosen that has the lowest p-value for a test statistic that assesses all parameters (i.e., MOS coefficients) simultaneously. The tests used here are permutation tests for independence using a quadratic form as the test statistic (see Hothorn et al. 2006, 2008). We have amended the text to include this information and the additional references.

*Section 2.2, Step 3: Which are the stopping criteria for the growth of the tree? What is the minimum sample size at a leaf node in the experiments reported in this paper (particularly in those experiments where a relatively small dataset is used)?*

We use the same hyperparameter values as the distributional forests of Schlosser et al. (2019). This means that nodes must have a sample size of at least 50 in order to be split again (`minsplit` = 50) and that terminal nodes (leaves) must have a sample size of at least 20 (`minbucket` = 20). The same values are used for all of the experiments, including those with only 3 years of training data. We have added this information to Sec. 3.2.1, where specific model setups for the precipitation forecasting are documented.

*Equations 6 and 7: To my understanding, these equations describe a type of local regression in which distance is measured based on the number of times two given data points belong to the same category in the different trees of a given forest. So the distance is specific to the problem at stake. This characteristic distinguishes this method from other methods that use random forests for postprocessing in the sense that $\theta$ is not directly given by the forest, but the forest provides a way to detect predictors that are close to the current predictors, and based on these neighbor predictors, a new set of parameters can be obtained (by retraining the model using only these weighted neighbors).*

Yes, this interpretation is correct. In forests with very simple "models" in the leaves (e.g., just a mean response or a success proportion) both approaches are equivalent. Thus, simply averaging mean responses from trees in a forest yields the same prediction as computing a weighted response based on the neighborhood weights. However, for parameters from more complex models (e.g., MOS coefficients or distributional parameters) the two approaches differ somewhat and the approach based on neighborhood weights is used more often in the literature.

*Based on this, I wonder:*

*What would be the performance of the proposed technique if the parameters θ provided by the forest were used directly for the postprocessing of the forecast (i.e., what is the impact of the neighbor approach on the performance of the method)?*

Thanks for the suggestion. We have compared the two approaches now: direct averaging of the predicted parameters from the individual trees vs. re-estimation of the base model using the neighborhood weights. The results are displayed in Figure 1 below. Both methods perform virtually identical when the full 24 years of training data are available. But when only 3 years of data are available the neighborhood weights perform slightly better. We have now mentioned these new results in the text.

Figure 1: CRPSS of neighbor method compared to averaging method for 3 and 24 years of training data.

*What would be the performance of the method if the distance metric were replaced by the classical Euclidean norm (like in the classical nearest neighbors approach)?*

The classical nearest neighbor approach ($k$-NN) with a Euclidean norm could make sense for a distributional forest, since the terminal nodes of each individual distributional tree would be expected to contain similar values for at least some of the predictors. The approach is less suited for a MOS forest, where terminal nodes instead correspond to separate MOS models and thus require a larger range of values for (certain) predictor variable(s). Another point that would need to be considered is that the Euclidean norm is not ideal for dealing with a high dimensional predictor space. Subsequently, some form of dimension reduction would likely be required for our application.

*What is the variability of the weights? Particularly in the small training sample cases. If the weight variance is not too high in the small training sample scenarios, then this may help to increase the robustness of the method because model parameters would be trained with a relatively larger sample than in the other methods. Is there a way in which this variability can be controlled and eventually tuned as a hyperparameter to maximize the performance of the method?*

Weights of a MOS forest are generally more variable than those of a distributional forest. Unsurprisingly, the average variance of an observation weight in the training data for a specific day in the test data (Fig. 2) is significantly larger for models trained on 3 years of data. This is because larger training data sets allow for more terminal nodes – given a fixed minimum node size – and thus generally have a larger fraction of weights equal to 0. The variability of the weights cannot be tuned directly, but can be influenced by changing the number of terminal nodes in a tree. For a given training dataset, this could be achieved by changing the minimum necessary size for a terminal node.

[Figure]

[Figure]

Figure 2: The average variance of the weights at Axams.

**Table 1: Could the authors elaborate more on why `tppow_mean` is excluded from the splitting variable list? It is not clear to me why that should be the case. Also, in the results section, the variable associated with the root split is the total column liquid condensate, which I assume is closely related to the precipitation rate (so the system is indirectly trying to use ttpow as a splitting variable)**

The ensemble mean of total precipitation `tppow_mean` is the "direct" weather prediction of our observations and contained within the base MOS model. It was excluded from the splitting variables to emphasize that MOS forests are able to account for additional (i.e., non "direct") variables during postprocessing. As you suspected, results are comparable when including `tppow_mean` among the splitting variables (Fig. 3), presumably due to correlations within the data.

[Figure]

Figure 3: The best performing method at each station when including `tppow_mean` among the MOS forest splitting variables.

***Table 1: Could the authors provide here or in the text some details about the configuration of the other methods? Since overfitting is a major concern when dealing with trees and forests, indicating the tree growth stopping criteria (or any other pruning approach) would be relevant for the comparison.***

By default, distributional forests use the same growth stopping criteria as those of the MOS forests, already mentioned above: `minsplit = 50` and `minbucket = 20`. For quantile regression forests, we use the default settings of the quantregForest package with a minimum terminal node size (`minbucket`) of 10. No pruning is performed.

***Figure 2: This figure is very interesting. However, I could not find cases in which precipitation occurred without being forecast (or maybe there is only one case in node 13). Is this because of the selected nodes, or is this a general property of the dataset?***

This is a general property of the RainTyrol dataset. Although approximately 42% of the observations (across all years and stations) are zero, only 0.7% of the ensemble mean forecasts are zero. Most likely this difference in frequencies results from the bilinear interpolation scheme used to generate ensemble forecasts for each station location and from the nature of ensembles themselves. For example, to obtain a forecast of no precipitation (i.e., `tppow_mean = 0`) would require that all ensemble members at each of the four neighboring gridpoints do not predict any precipitation.

***4.1: The names given to the different predictors are not clear. For example, what does `pwat_mean_max` mean? I assume the mean is from the ensemble mean, but I cannot interpret the max. This also applies to other names: `t500_sprd_min`, `tppow_sprd1824`.***

In addition to the ensemble mean of the 24 hour precipitation forecasts between +6h and +30h, the RainTyrol dataset also contains forecasts based on 6 hourly intervals. For example, the variable `tppow_sprd1824` is the spread of the 6h precipitation forecasts issued by the ensemble for the period between 18 and 24 UTC (i.e., lead times of +18h and +24h).

Variable names can contain two underscores. The expression after the first underscore indicates how forecasts are aggregated over the ensemble dimension (i.e., `sprd` for spread of the ensemble). The expression after the second underscore describes aggregation over the lead times within the 24h accumulation period (i.e., mean, maximum or minimum over +6h, +12h, +18h, +24h, +30h). Subsequently, `pwat_mean_max` is obtained by computing the mean of the ensemble forecast of `pwat` for each lead time, and then taking the maximum of these ensemble means.

We have added information about the variables and their naming convention to the paper.

***Regarding `tppow_sprd1824`, later in the text or in a figure caption, it is said that it corresponds to the spread over the 18–24 hour lead time period. Why did the authors choose this period to characterize the ensemble spread?***

The ensemble spread over the 18-24 hour period (`tppow_sprd1824`) is one of many variables contained within the RainTyrol dataset. Summer rainfall in Tirol is often caused by convection during the late afternoon or evening hours. Including forecasts from this time period can be valuable for postprocessing since NWP biases and miscalibrations may behave differently depending on the nature of the precipitation event. We have now mentioned this in the text.

*L300 "if the variable observed is not a direct output of the NWP model". This is unclear. Why can't physical quantities other than the ones observed be used to model the conditional probability distribution parameters?*

You are right that forecasts of physical quantities other than the ones observed could be used to model the conditional probability distribution parameters, but in that case it may be difficult to specify a suitable MOS regression that works well and is still physically meaningful or natural to understand. In order to avoid any confusion though, we have removed the highlighted phrase.

*L141 $\gamma_0$ is introduced here, but it has not been defined before ( $\sigma$ is used instead in the previous discussion).*

You are correct, we have now changed $\gamma_0$ to $\sigma$.

*Equation 7: The meaning of the denominator is not clear.*

Dividing by the size of the terminal node $| \mathcal{P}_p^t |$ avoids underrepresenting trees with more (and thus generally smaller) terminal nodes when calculating the weights. We have now mentioned this in the text.

*Equation 8: Please clarify the meaning of $\phi$ and $\Phi$.*

$\phi$ and $\Phi$ refer to the probability density function and cumulative density function of a standard Gaussian distribution, respectively. This has been added to the text.

*L225 rates are*

We believe the sentence is correct as is: "The first split of the tree separates rare ($n$ = 23) weather situations ..."

*Figure 3: Please clarify the meaning of the titles of the panels ("Location" and "Scale").*

The titles refer to the location and scale parameters of the response distribution (i.e., $\mu$ and $\sigma$, respectively). This has been clarified in the figure caption.

*In L268 and Caption Fig. 5, CRPS is used instead of CRPSS*

We have changed CRPS to CRPSS at L268. The caption of Fig. 5 was technically correct since the circle sizes corresponded to differences in CRPS not CRPSS. For consistency, we have now modified the figure to instead show the CRPSS of the best method with respect to the second best method. Subsequently, new cutoffs for the circle size are approximately one order of magnitude larger.

---

## Author Comment (AC2)

**Responses to Reviewer 2**

*This manuscript introduces a new type of postprocessing of numerical weather forecast using MOS random forests. It clearly describes the methodology, highlighting both the advantages and limitations. While the structure of the paper is closed to Schlosser et al. (2019), I consider the manuscript contains enough new results to be published in Nonlinear Processes in Geophysics. The manuscript can be considered as an update of Schlosser et al. (2019). I therefore recommend publication after minor revisions. Please find my specific comments and technical corrections below.*

Thank you for taking the time to read our manuscript and for your constructive remarks. Please find point-by-point responses to your specific comments below.

*Line 23: What is the meaning of "homogeneous" here? Please elaborate.*

Here, homogeneous means that a single MOS model with constant coefficients can be used to effectively postprocess the forecast. That is, the systematic biases and miscalibrations of the numerical weather model are relatively constant within the dataset. We have clarified this in the text.

*Line 37: Please cite some references of random forests used to perform ML-based postprocessing.*

Three references are found in the second half of the sentence. We have now added a fourth reference as well.

*Lines 107-108: What is the source of the citation, if it's a citation?*

This is not a citation and the quotations were only used for emphasis. To avoid confusion, we have removed the quotations and instead italicized the phrase.

*Line 179: It should be added that July 19, 2011 is missing for all the 95 stations in package RainTyrol (version: 0.2-0, date: 2020-01-13). This might not affect the results, but it is important to mention about it.*

Thanks for noticing this! The missing day is in the last four years of the dataset that were used for model evaluation. Naturally no forecasts were made or evaluated for this day. We have mentioned the missing day in the text.

*Lines 239: Please summarize the physical meaning of this, i.e., the mechanism, rather than only listing the variables. Or is it due to GEFS to generate more days with small amount of rainfall than observed? (which seems to be suggested by the authors.)*

It is difficult to give a physical explanation for the particular MOS model in this terminal node. This is also not our objective since ultimately the postprocessed forecasts are obtained from a MOS forest and not the individual tree. Our goal here is to highlight the statistical interpretation of a MOS tree's terminal node: that is, to explain how a tree outputs a particular MOS model for the predictor values corresponding to the preceeding splits. We have now adjusted the text to emphasize this distinction.

*Lines 264-267 and Figure 5: This is an interesting result. Particularly compared to Fig. 8 of Schlosser et al. (2019), which shows a less organized spatial distribution of best postprocessing. But the question of why the NE-SW distribution is not really discussed in the main text. Is it solely due to the topography? From the figure, it seems the terrain is lower to the NE and higher to the SW. Also, as the MOS random forest is weather adaptative, could this spatial distribution be linked to the main mode of variability of weather in July? (either in the real world or in the GEFS world). It would be interesting to discuss this possibility in the manuscript.*

When compared to Fig. 8 of Schlosser et al. (2019), our results show a much more organized spatial distribution of the best postprocessing method. This is mainly because we also compared our method to quantile regression forests, which do not assume a parametric response distribution. In contrast, Schlosser et al. (2019) compared various methods that all assume the same parametric distribution. It seems that this assumption does not hold as well for the southern and western regions where the terrain is higher. We have now mentioned this in the revision.

MOS forests are not the only weather-adaptive model we compared: both the distributional forests and quantile

regression forests are also weather-adaptive. It is difficult to link the observed spatial distribution to a certain mode of weather variability in July since we cannot compare our results to any other months.

*Line 288: "new stations (or measurement instruments) are installed all the time". Is "new" here equivalent to "additional" or to "in replacement of"? If it is additional, is it to document highler altitudes in the case of complex terrain, knowing the costs of installation and maintenance? Is it true at the globe scale? I am not sure this assertion is necessary (and accurate) here. On the other hands, if this correct, it might in fact introduce more biases in dataset (instrument drift, error in transcription, system failure...). Please elaborate.*

The "new" here refers to either additional stations or modifications to existing stations (e.g., different sensors, orientation, changing surroundings, etc.). You are exactly correct that including such changes can introduce biases in the dataset. Subsequently, there is very little data avaiable for new stations and these greatly benefit from a *robust* postprocessing method such as MOS forests.

*Line 289: How would you interpret this citation in the context of this study? Does it suggest that because the postprocessing is weather adaptive, it is constrained by the model's world weather?*

The citation "data samples containing numerical model output are a perishable commodity" means that the datasets available for training postprocessing models often have a limited size. Weather-adaptive postprocessing methods are able to take into account many predictors during postprocessing, but therefore require more data for training. The advantage of MOS forests is robustness: the ability to account for these additional variables and issue skilled forecasts even when the data size is limited.

*This study only focuses on July. It would be interested to see the robustness on this approach in other seasons, because the regional influence (main modes of variability) vary along the year. And could be linked to the comment on Fig. 5, i.e., if we see a change in the spatial distribution of best postprocessing.*

We are sticking to the July data contained within RainTyrol to allow for a better comparison with Schlosser et al. (2019), who also considered weather-adaptive postprocessing based on generalized additive models for location, scale, and shape (GAMLSS). These GAMLSS models incorporate additional predictors into postprocessing using variable selection based on prior knowledge or with boosting.

*From this, another question would be: Is it possible to objectively define the most appropriate postprocessing? From the main mode of weather variability? Please discuss the possible directions.*

We are not sure what this question entails exactly. Do you mean whether it would be possible to establish a kind of "meta learner" which predicts which learner performs best for a certain setup (e.g., based on topography and climatology)?

If so, we think that this might be possible but is probably not straightforward. So far we have always performed some form of benchmark/validation study, possibly in combination with prior expertise, to select the best learner/model for a given area of application.

*Throughout the manuscript, the author sometimes use the term "MOS forests" (for example l. 12 or l. 55) and sometimes the term "MOS random forests" (l. 43 for example). Please review the whole manuscript to homogeneize the terms (and maybe use an acronym).*

We have kept the full name of the method "MOS random forests" in the title and the abstract, but used the abbreviation "MOS forests" in the body of the manuscript. Use of the abbreviation is now explicitly addressed the first time the method is mentioned in the introduction (second to last paragraph).

*Line 8: "ML" is not previously defined.*

We have written out "machine learning" and dropped the abbreviation.

*Line 95: Maybe write "...the postprocessing literarure is the nonhomogeneous Gaussian ...".*

Thanks, we have adjusted the sentence for improved readability.

*Line 107: the sentence looks incomplete ("...a single MOS tree partitions the predictor space ...").*

We think this is a complete sentence. The word "partitions" functions as a verb.

*Line 180: I would a short sentence to tell how this number is defined, i.e., "median of all estimated power coefficient (Stauffer et al. 2017a)".*

This has been added.

*Line 184: It would be easier for the readers to mention that the authors are specifically referring to Table 1 of Schlosser et al. (2019).*

Yes, that is true. We have now explicitly referenced Table 1.

*Line 221: Please indicate where is the station of Axams located in Figure 5 (preferred). Or at least refer to Fig. 8 of Schlosser et al. (2019).*

We have now referred to Fig. 8 of Schlosser et al. (2019).

*Line 270: It would be better to define "PIT" and "PIT histograms" here.*

You're right. We have added the full name "probability integral transform" and a noted that uniform histograms indicate good calibration.

*Line 283: "very little data". Do the authors mean small sample size?*

Yes, we have adjusted the text.

*Figure 2: A "l" is missing in "Dashed and solid lines ..."*

This has been corrected.

*Figure 3: What is the meaning of "Location" and "Scale"?*

Location and scale refer to the two distributional parameters ($\mu$ and $\sigma$, respectively). This has been clarified in the caption.

*Figure 5: Shouldn't it be "CRPSS"? Also, the background is not described in the caption. And the size of the circles should be included in the legend.*

We have modified the figure to show the CRPSS with respect to the second best method, rather than differences in CRPS, as was originally done. The size of the circles is now also included in the figure and the background shading described in the caption.

---

## Author Response (AR2)

Dear Dr. Miyoshi,

thank you for taking the time to read our manuscript and for your positive comments. Upon reviewing the text, we found four typos and have corrected these for the new submission.

Best regards,

The authors